# Antagonistic relationship of NuA4 with the non-homologous end-joining machinery at DNA damage sites

**Salar Ahmad, Valérie Côté, Xue Cheng, Gaëlle Bourriquen, Vasileia Sapountzi, Mohammed Altaf¤, Jacques Côté***

St-Patrick Research Group in Basic Oncology, Centre Hospitalier Universitaire de Québec-Université Laval Research Center, Laval University Cancer Research Center, Quebec City, Canada

¤ Current address: Chromatin and Epigenetics Lab, Department of Biotechnology, University of Kashmir, Srinagar, India
* jacques.cote@crhdq.ulaval.ca

**Data Availability Statement:** All relevant data are within the manuscript and its Supporting Information files.

## Abstract

The NuA4 histone acetyltransferase complex, apart from its known role in gene regulation, has also been directly implicated in the repair of DNA double-strand breaks (DSBs), favoring homologous recombination (HR) in S/G2 during the cell cycle. Here, we investigate the antagonistic relationship of NuA4 with non-homologous end joining (NHEJ) factors. We show that budding yeast Rad9, the 53BP1 ortholog, can inhibit NuA4 acetyltransferase activity when bound to chromatin *in vitro*. While we previously reported that NuA4 is recruited at DSBs during the S/G2 phase, we can also detect its recruitment in G1 when genes for Rad9 and NHEJ factors Yku80 and Nej1 are mutated. This is accompanied with the binding of single-strand DNA binding protein RPA and Rad52, indicating DNA end resection in G1 as well as recruitment of the HR machinery. This NuA4 recruitment to DSBs in G1 depends on Mre11-Rad50-Xrs2 (MRX) and Lcd1/Ddc2 and is linked to the hyper-resection phenotype of NHEJ mutants. It also implicates NuA4 in the resection-based single-strand annealing (SSA) repair pathway along Rad52. Interestingly, we identified two novel non-histone acetylation targets of NuA4, Nej1 and Yku80. Acetyl-mimicking mutant of Nej1 inhibits repair of DNA breaks by NHEJ, decreases its interaction with other core NHEJ factors such as Yku80 and Lif1 and favors end resection. Altogether, these results establish a strong reciprocal antagonistic regulatory function of NuA4 and NHEJ factors in repair pathway choice and suggests a role of NuA4 in alternative repair mechanisms in situations where some DNA-end resection can occur in G1.

## Author summary

DNA double-strand breaks (DSBs) are one of the most harmful form of DNA damage. Cells employ two major repair pathways to resolve DSBs: Homologous Recombination (HR) and Non-Homologous End Joining (NHEJ). Here we wanted to dissect further the role played by the NuA4 (Nucleosome acetyltransferase of histone H4) complex in the

**Funding:** This work was supported by a grant from the Canadian Institutes of Health Research (cihr-irsc.gc.ca) to J.C. (FDN-143314). S.A. held a Desjardins/Fondation du CHU de Québec (fondationduchudequebec.org) studentship and X. C. a Fonds de Recherche du Québec-Santé (www. frqs.gouv.qc.ca) studentship. J.C. holds the Canada Research Chair in Chromatin Biology and Molecular Epigenetics (www.chairs-chaires.gc.ca). The funders had no role in study design, data collection and analysis, decision to publish, or preparation of the manuscript.

**Competing interests:** The authors have declared that no competing interests exist.

repair of DSBs. Budding yeast NuA4 complex, like its mammalian homolog TIP60 complex, has been shown to favor repair by HR. Here, we show that indeed budding yeast NuA4 and components of the NHEJ repair pathway share an antagonistic relationship. Deletion of NHEJ components enables increased recruitment of NuA4 in the vicinity of DSBs, possible through two independent mechanisms, where NuA4 favors the end resection process which implicates it in repair by single-strand annealing (SSA), an alternate homology-based repair pathway. Additionally, we also present two NHEJ core components as new targets of NuA4 acetyltransferase activity and suggest that these acetylation events can disassemble the NHEJ repair complex from DSBs, favoring repair by HR. Our study demonstrates the importance of NuA4 in the modulation of DSB repair pathway choice.

## Introduction

The genomic integrity of a cell is under constant assault from various endogenous and exogenous sources. One of the most deleterious manifestation is DNA double strand breaks (DSBs), as failure to repair a DSB can lead to loss of genetic information and be an underlying cause for tumor progression in higher eukaryotes [1,2]. Cells have two main pathways to repair DSBs. First, Homologous Recombination (HR) is carried out in the S/G2 phase of the cell cycle, relies on 5'-3' resection of the broken DNA ends and uses the homology on the sister chromatid as a template for repair [3]. Second, Non-Homologous End Joining (NHEJ) is carried out throughout the cell cycle but it is the dominant form of repair in the G1 phase. NHEJ may or may not use processing of the broken ends before ligation, making it a more error-prone pathway than HR [4]. Cells also have backup repair pathways such as Single Strand Annealing (SSA) and alt-NHEJ, which share some components with HR and NHEJ, respectively. These pathways come into play under special conditions when the main repair pathways are hindered due to genetic mutations, absence of sister chromatid and state of the cell cycle [5,6].

The yeast NuA4 histone acetyltransferase complex is essential for cell viability, is composed of 13 subunits and is homologous to the mammalian TIP60/p400 complex [7,8]. NuA4/TIP60 acetylates histones H4 and H2A N-terminal tails in chromatin, as well as variants H2A.X/H2A.Z [9–13]. NuA4/TIP60 has been well documented for its role in transcription regulation, at the initiation and elongation steps [14,15]. NuA4 is also known to acetylate non-histone proteins and regulate a variety of cellular processes such as autophagy and gluconeogenesis [16,17]. Temperature sensitive mutants of Esa1, the yeast catalytic subunit, as well as point mutants of H4 and H2A target lysine residues show growth defects in presence of various DNA damaging chemical agents, underpinning the importance of NuA4 and acetylation for the repair of DNA breaks [7,9,18–20]. NuA4 has been shown to play an important role for the recruitment of chromatin remodelers such as INO80, SWI/SNF and RSC to DNA repair sites [20–22]. We have shown that NuA4 is rapidly recruited to a DSB by the MRX complex (Mre11-Rad50-Xrs2) and this is most likely through a DNA damage-induced phospho-dependent interaction with the Xrs2 protein [23]. Very recently, we found that NuA4 is recruited to DSBs along another important HAT complex, SAGA, and their combined action is essential for DNA end resection to occur and HR [24]. In addition, we have shown that NuA4 directly acetylates RPA to modulates its dynamic association with ssDNA during the repair process [23]. NuA4 has also been implicated in opposing Rad9 checkpoint activity after DNA damage in budding yeast [10,25]. This is paralleled by the mammalian TIP60 complex which was

shown to antagonize 53BP1 (homolog of yeast Rad9) in the vicinity of DSBs to favor repair by HR [26,27]. Tip60-dependant acetylation and activation of ATM has also been argued to be involved in DNA damage signaling [28].

In this study we dissected the functional relationship of NuA4 with the NHEJ repair pathway. We show that, upon removal of Rad9 and NHEJ factors, NuA4 is strongly recruited to DSBs during the G1 phase of the cell cycle, alongside DNA end resection, in a MRX- and Lcd1/Ddc2-dependent manner. Conditions of defective NuA4 acetyltransferase activity decrease the hyper-resection phenotype of NHEJ mutants, implicating NuA4 in the resection-based SSA repair pathway in G1. In addition, NHEJ core factors Nej1 and Yku80 are targets of NuA4-dependent acetylation which lowers NHEJ efficiency, disrupts physical interactions and increases resection. Our results support a model in which NuA4 and the NHEJ machinery are negatively regulating each other, NuA4 favoring repair of DSBs by resection-based pathways in part through inhibition of NHEJ factors.

## Results

### Deletion of NHEJ factors leads to recruitment of NuA4 at DSBs in G1

Yeast Rad9, similar to its mammalian homolog 53BP1, is known to slow down or to keep a check on the rate of DNA end resection [29–32]. 53BP1 has been shown to have an antagonistic relationship with mammalian NuA4/TIP60 [27]. They compete to bind the H4K20me mark and target H2AK15 modifications, implicating them in the selection of the DSB repair pathway [26]. In yeast, NuA4 mutant cells accumulate in the G2 phase at non-permissive temperature, in a Rad9-dependent manner [10]. In addition, NuA4 mutants defective in acetylation show a strongly increased Rad9-dependent G1 checkpoint upon DNA damage, implicating NuA4 in an antagonistic relationship with Rad9 during the DNA damage response [25].

Yeast Rad9 binds chromatin near DSBs through bivalent interaction with two histone marks, H3K79me by its Tudor domain and H2AS129ph by its BRCT domain (γH2A) [33–35]. To determine if Rad9 binding to chromatin can affect NuA4 activity (**Fig 1A**), we performed *in vitro* histone acetyltransferase (HAT) assays using native chromatin purified from cells treated with a DNA damaging agent, MMS. Wild type and Tudor or BRCT mutant recombinant Rad9 (aa639-1308, **S1A Fig**) were incubated with chromatin and purified native NuA4 complex, followed by HAT assay with radioactive Acetyl-CoA. Wild type Rad9 was able to significantly block acetylation of nucleosomes by NuA4, whereas Rad9-Y798Q and Rad9-K1088E were significantly less efficient at doing so, reflecting their decreased ability to bind chromatin (**Figs 1B and S1B**).

We and others have shown that NuA4 is recruited to DSBs mainly during the S/G2 phase of the cell cycle [23,36]. To study whether Rad9 may affect recruitment of NuA4 at DSBs *in vivo*, we deleted *RAD9* and performed chromatin immunoprecipitations (ChIP)-qPCR at the HO-induced DSB in asynchronous and G1-arrested cells. It is important to indicate that all the ChIP experiments presented in this study were controlled for consistent level of HO-induced DSB formation (**S1E–S1I Fig**) as well as tight synchronization of cells in G1 (*bar1* mutant background, FACS and microscopic analysis). While asynchronous cells showed no significant difference in NuA4 binding between wild type and *RAD9* deletion mutant (**S1C Fig**), G1 synchronized cells showed a small but significant increase in the recruitment of NuA4 (**Fig 1C**), along with the expected increase of DNA end resection as measured with Rfa1, the largest subunit of RPA (**Fig 1D**).

Deletion of yeast Ku (Yku) in asynchronous cells did not significantly affect recruitment of NuA4 at the HO-induced DSB [23]. However, cells synchronized in G1 phase showed strong

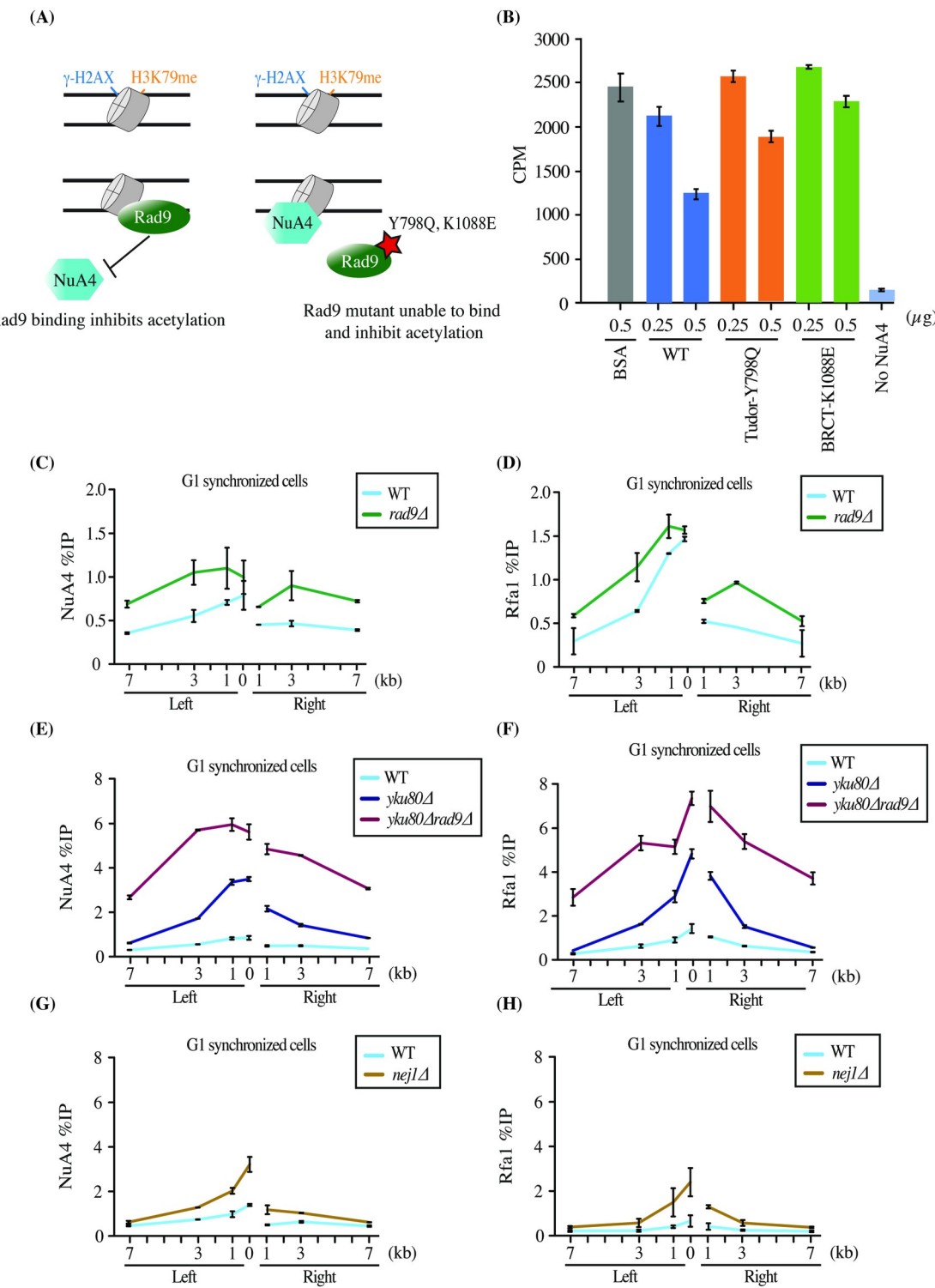

**Fig 1. NHEJ factors inhibit NuA4 recruitment and end resection at a DSB in G1.** (A) Schematic representation of the experiment in (B). Wild type Rad9 binds to chromatin and inhibits NuA4 from acetylating histones, whereas Rad9 mutants are unable to bind chromatin efficiently, thus resulting in acetylation of histones by NuA4. (B) *In vitro* HAT assay showing inhibition of NuA4 acetyltransferase activity by Rad9 using yeast native chromatin purified from MMS-treated cells and recombinant Rad9 (wild type, tudor-Y798Q and BRCT- K1088E domain mutant, S1A Fig). Recombinant Rad9 wild type and mutants were incubated with 500ng of chromatin. Afterwards NuA4 was added along with $^{3}$H-acetyl-CoA followed by spotting on P81 membrane and scintillation counting. Error bars indicate the ranges of duplicate reactions. Only 0.5ug of WT and tudor mutant show a statistically

significant decrease of acetylation compared to control (BSA). Higher amount of Rad9 leads to more inhibition (S1B Fig). (C-H) ChIP-qPCR showing increased recruitment of NuA4 (Eaf1 subunit) and Rfa1 (RPA) in *rad9Δ* (C-D), y*ku80Δ*, y*ku80Δrad9Δ* (E-F) and *nej1Δ* (G-H) compared to wild type at an HO-induced DSB in G1. ChIP-qPCR analysis was performed using Eaf1 and Rfa1 antibodies. Cells with *bar1Δ* background were synchronized in G1 phase using α-factor for 6 hours in YP-Raffinose followed by addition of galactose for 3 hours to induce DSB. The data represents two independent biological replicates and error bar denotes range between the replicates. Yeast strains used had similar HO cutting efficiency (S1E–S1I Fig).

recruitment of NuA4 at the DSB in the absence of Yku80 (**Fig 1E**), reminiscent of a reported increased binding of some proteins shared by RPD3S, SWR-C and NuA4 complexes in the absence of Yku70 [36]. ChIPs with Rfa1 also showed strongly increased signal for RPA around the DSB site, indicating abnormal high DNA end resection in G1 (**Fig 1F**). We wanted to know if this antagonistic relationship was exclusive for Yku or extended to other NHEJ factors. Deletion of *NEJ1*, a core NHEJ factor, in G1 synchronized cells resulted again in increased NuA4 recruitment around the HO DSB (**Fig 1G**)**,** as well as higher resection (**Fig 1H**), which is not observed in asynchronous cells (**S1D Fig**).

In parallel, *RAD9* was deleted in the *YKU80* deleted background. Previous studies have shown that the double deletion of *RAD9* and *YKU70* results in hyper-resection of DNA ends at DSBs in the G1 phase of the cell cycle [37]. We confirmed this finding as indicated by a very strong increase of Rfa1 signal seen in *rad9Δyku80Δ* cells (**Fig 1F**). Interestingly, a similar effect was observed for NuA4, with the double deletion mutant giving a much stronger signal compared to single y*ku80Δ* cells (**Fig 1E**). Altogether, these data demonstrate that the NuA4 complex can be efficiently recruited at a DSB in G1 in the absence of core NHEJ factors, Yku80 and Nej1, and also with the deletion of *RAD9*. This supports the concept of an antagonistic relationship between NuA4 and the NHEJ machinery for repair of DNA breaks.

## Xrs2-dependent recruitment of NuA4 at DSB in G1

NuA4 subunit Tra1 shows high domain homology with Tel1/ Mec1, members of the PIKK family. Both Tel1 and Mec1, like their mammalian counterpart ataxia telangiectasia mutated (ATM), AT-related (ATR) and DNA-dependent protein kinase (DNA-PK), are important for signaling of the DSB and recruitment of downstream repair proteins [38,39]. We have recently shown that Tra1 gets phosphorylated upon DNA damage and its C-terminus containing the PI3K-like domain can directly bind Xrs2 *in vitro*, an interaction likely involved in the recruitment of NuA4 at the HO induced DSB site [23]. To understand if Xrs2 is required for NuA4 recruitment at the DSB in G1, we performed ChIP-qPCR in *XRS2* deletion background. As seen in (**Fig 2A**), the NuA4 signal detected in G1 in the absence of Rad9 was completely lost when *XRS2* is deleted. The same can be said for end resection measured by Rfa1/RPA (**Fig 2B**). Strikingly, deletion of *XRS2* in the background with hyper-resection (*rad9Δyku80Δ*) also abrogated NuA4 recruitment and resection in G1 cells (**Figs 2C, 2D** and **S2D**). We have previously shown that NuA4 is recruited through a two-step mechanism at DSBs in S/G2: first, by the MRX complex, through a phospho-dependent interaction with Xrs2; and second, by spreading on each side of the break along DNA end resection [23]. Since deletion of *XRS2* blocks resection, it was unclear whether the loss of NuA4 recruitment is solely due to the loss of interaction with Xrs2 or also partly due to the loss of resection itself. The *mre11-H125N* nuclease-defective mutant background was used to partly address this issue since it is known to slow down the rate of resection in G1 synchronized cells [40]. As shown in **Fig 2E and 2F**, while there was a decrease in the Rfa1 signal close to the break, NuA4 detection seemed unaffected. These results suggest that NuA4 recruitment at a DSB in G1 requires the Xrs2 subunit of the Mre11/Rad50/Xrs2 (MRX) complex, in a resection-independent manner.

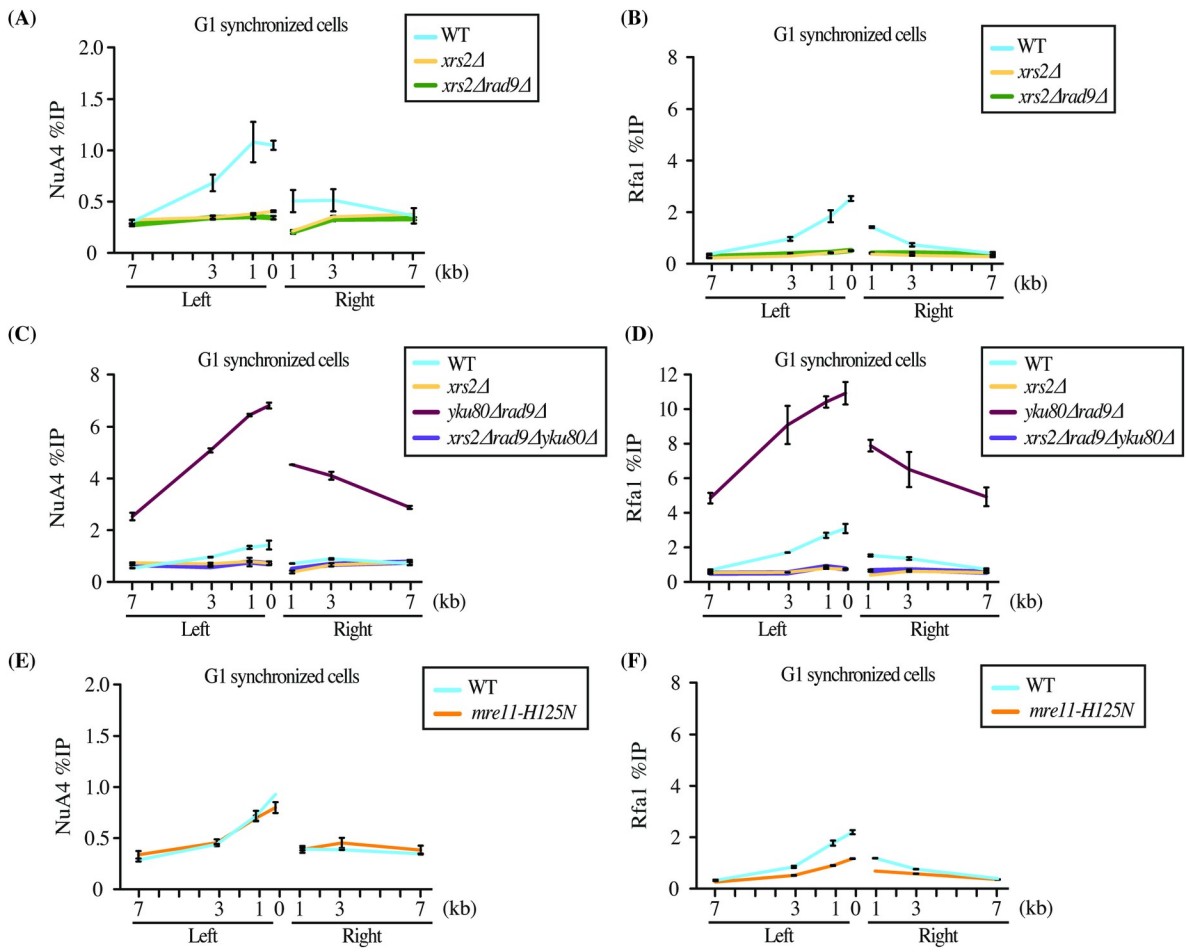

**Fig 2. Xrs2-dependent recruitment of NuA4 at a DSB in G1.** (A-D) ChIP-qPCR of Rfa1 and Eaf1 shows loss of NuA4 (A, C) and resection (B, D) in *xrs2Δ*, *xrs2Δrad9Δ* and *xrs2Δrad9Δyku80Δ* compared to wild type at HO-induced DSB in G1 synchronized cells. See also S2D Fig for direct measurement of DNA end resection by qPCR. (E-F) ChIP-qPCR in the *mre11-H125N* mutant strain shows decreased resection but unaffected NuA4 recruitment compared to wild type in G1 cells. ChIP-qPCR was performed as described in (Fig 1). Yeast strains used showed similar HO cutting efficiency (S2A–S2C Fig).

## Lcd1/Ddc2-dependent recruitment of NuA4 at DSBs

Xrs2 harbors the Nuclear Localization Sequence (NLS) required for Mre11 import into the nucleus. Thus, loss of Xrs2 results in failed assembly of the MRX complex at the DSB [41]. To overcome this issue, we used a low copy number plasmid expressing Mre11 fused to a NLS (referred to as *MRE11-NLS*). In accordance with previous studies, expression of Mre11-NLS suppressed the growth defect of *xrs2Δ* cells in the presence of a DNA damaging agent (MMS, **S3A Fig**) [41,42]. Interestingly, expression of Mre11-NLS in G1 cells carrying the double deletion *rad9Δyku80Δ* in addition to *xrs2Δ* led to a significant increase in NuA4 recruitment and resection (**Fig 3A and 3B**). Since the suppression of *xrs2Δ* by Mre11-NLS is known to be partial, we also tested the Mre11-NLS-X85 fusion which adds the C-terminal 85aa Tel1-interacting domain of Xrs2 to Mre11 [43] (distinct from the N-terminal FHA-BRCT NuA4-interacting domain [23]). This construct led to an even much stronger increase of NuA4 recruitment and resection at the DSB in the absence of Xrs2, Rad9 and Yku80 (**Fig 3C and 3D**). These results demonstrate the existence of an Xrs2-independent mode of NuA4 recruitment at DSBs. Importantly, this Xrs2-independent mechanism is not restricted to cells

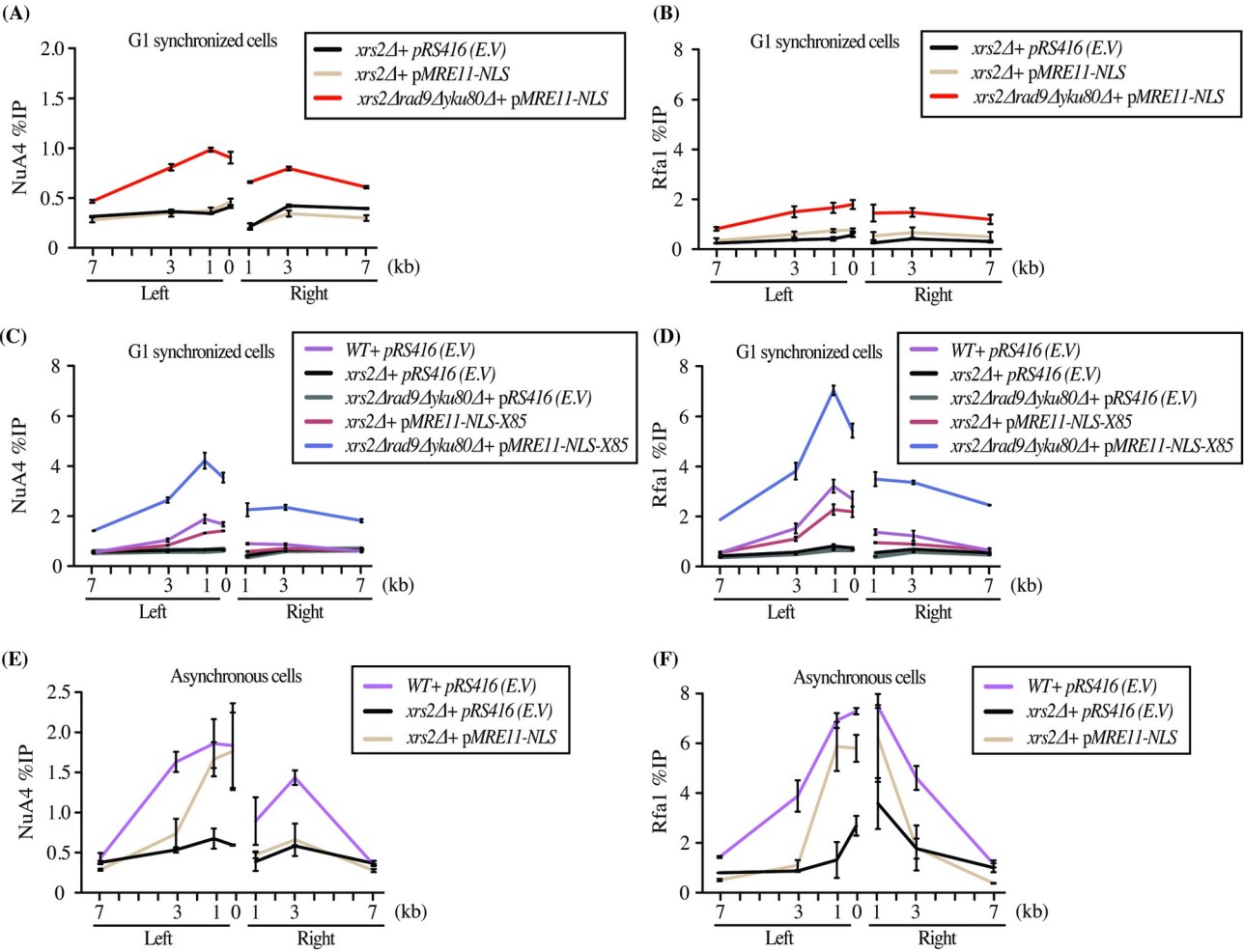

**Fig 3. Alternate mode of NuA4 recruitment in G1 in conditions of Xrs2 functional bypass.** (A-B) Resection and NuA4 recruitment is partially restored in the *xrs2Δ rad9Δyku80Δ* background with expression of Mre11-NLS from a low copy number ARS:CEN plasmid. ChIP-qPCR of Eaf1 and Rfa1 was performed as in (Fig 1) in cells synchronized in G1. (C-D) Resection and NuA4 recruitment is largely restored in *xrs2Δ rad9Δyku80Δ* background by expression of Mre11-NLS-X85 that more fully bypasses Xrs2 functions. (E-F) Expression of Mre11-NLS also partially rescues NuA4 recruitment and resection (Rfa1 signal) in the *xrs2Δ* background in asynchronous cells (on the left side proximal to the DSB site). (See S3 Fig for *xrs2Δ* phenotypic suppression by Mre11-NLS and HO-cutting efficiencies).

in G1 as asynchronous cells also show a rescue for NuA4 recruitment and resection with the introduction of *MRE11-NLS* in *xrs2Δ* cells (**Fig 3E and 3F**).

We previously reported that the Tra1 subunit of NuA4 can interact directly *in vitro* with Lcd1/Ddc2 (ATRIP in mammals) [23]. Lcd1/Ddc2 is the binding partner of Mec1 (ATR in mammals) and is responsible for the recruitment of Mec1 onto the RPA-coated ssDNA generated by resection [44]. Importantly, we have previously shown that deletion of *LCD1/DDC2* did not significantly affect the recruitment of NuA4 at the HO break in asynchronous cells [23](see also **S4B Fig**). To establish whether NuA4 interaction with Lcd1/Ddc2 could be in fact an alternate mechanism of NuA4 recruitment when Xrs2 function is genetically bypassed (as above), we deleted *LCD1* (in a *SML1* deleted background to overcome the synthetic lethality) [45,46]. However, the introduction of *MRE11-NLS* in this background is insufficient to rescue resection, making conclusions about NuA4 recruitment difficult (**S4A and S4B Fig**). To overcome this issue, we overexpressed long range resection exonuclease Exo1. Expression of

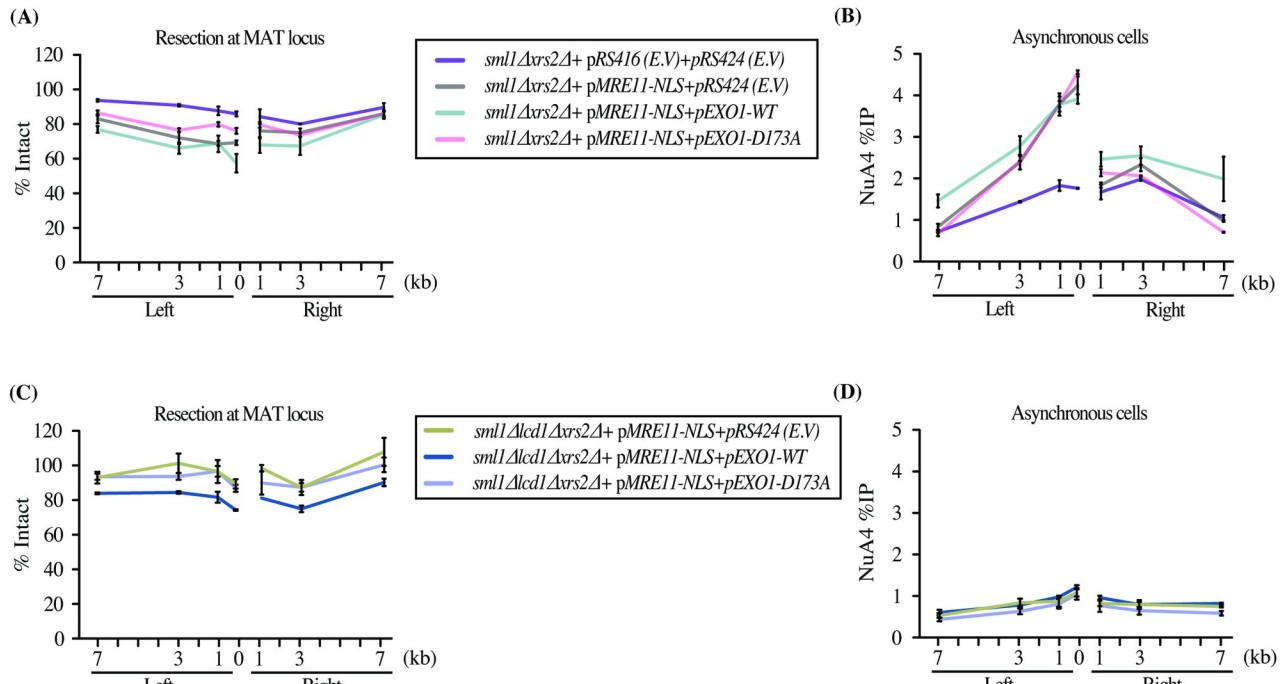

**Fig 4. The alternate mode of NuA4 recruitment at the DSB is Lcd1/Ddc2-dependent.** (A-B) Resection and NuA4 recruitment can be further rescued in the *xrs2Δ* background by expressing Mre11-NLS along catalytically active Exo1. End resection is directly measured by decreased DNA signal near the break and NuA4 recruitment is monitored as before. Expression of a catalytically dead Exo1 is used as control (Exo1-D173A). (C-D) Resection is also rescued in the *lcd1Δxrs2Δ* background with *pMRE11-NLS* and overexpression of Exo1-WT but not Exo1-D173A mutant, but this is not the case for NuA4 recruitment. NuA4 signal does not show a resection-correlated increase with Exo1-WT compared to the Exo1-D173A mutant or the empty vector. Resection is presented as % of Intact DNA determined by qPCR on genomic DNA at indicated sites and normalized to the negative control intergenic V. Cells were grown in YP-Raff till early log phase followed by addition of galactose for 3 hours to induce the DSB. Yeast strains used had similar HO cutting efficiency (S4C and S4D Fig).

Mre11-NLS along with Exo1 overexpression led to diminished DNA signals on each side of the break after HO induction, indicating DNA end resection (**Fig 4A and 4C**). This is dependent on Exo1 activity since a nuclease dead mutant of the protein (Exo1-D173A) behaves like the empty vector control (EV). In conditions of *XRS2* deletion partly suppressed by Mre11-NLS, the expression of active Exo1 led to a further increase of NuA4 recruitment (**Fig 4B**). However, when *LCD1/DDC2* was deleted in these conditions that allow significant DNA end resection (**Fig 4C**), NuA4 signals failed to increase (**Fig 4D**). Therefore, these results indicate that, in conditions in which Xrs2 function is genetically bypassed, there is an alternate Lcd1/Ddc2-dependent mechanism of NuA4 recruitment at the DNA break, likely through a direct physical interaction.

## NuA4-dependent acetylation is important for hyper-resection in G1 cells and repair by the SSA pathway

DSBs flanked by direct repeats are repaired by a Rad52-dependent pathway called single strand annealing (SSA). The resected ends anneal at the repeats and the intervening DNA along with one repeat are removed by a nucleolytic process [47]. It has been well documented that Rad52 is essential for repair of DSBs by SSA in G1 cells [37]. To investigate NuA4 possible function in such process, we performed ChIP-qPCR to measure Rad52 recruitment to DSBs in G1 and our mutant backgrounds. As expected in the hyper-resection *rad9Δyku80Δ* mutant background, very high recruitment of Rad52 was detected in G1 cells (**Fig 5A**). This was also seen

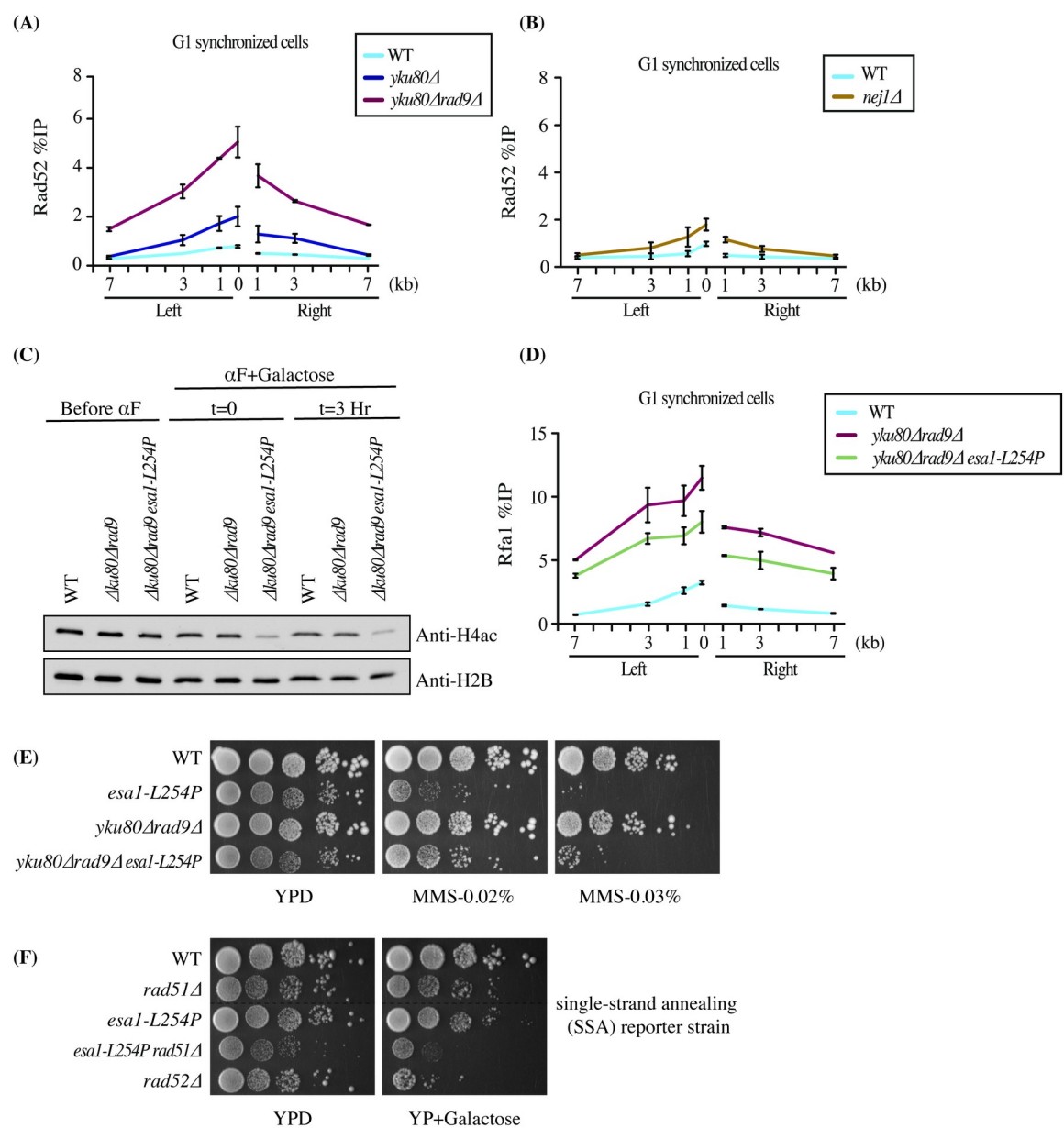

**Fig 5. Decreased hyper-resection in a NuA4 acetylation defective mutant in G1 and defect in DNA break repair by SSA.** (A-B) ChIP-qPCR of Rad52 around the HO-induced DSB in G1 blocked cells. *rad9Δyku80Δ*, y*ku80Δ* (A) and *nej1Δ* (B) show higher recruitment of Rad52 compared to wild type conditions. (C) Reduced histone H4 acetylation in the *esa1-ts* background at semi-restrictive temperature. Western blot analysis of WCE from yeast strains used for ChIP-qPCR in (D). Samples were collected before addition of α-factor at permissive temperature (23˚C) in YP-Raff. α-factor was added and cultures were moved to semi-restrictive temperature (32˚C) followed by addition of galactose to induced the HO cut for 3 hours. (D) ChIP-qPCR of Rfa1 around the HO-induced DSB in G1 cells showing reduced resection in the *rad9Δyku80Δ* background with the introduction of the NuA4 *esa1-ts* defective allele. Yeast strains used had similar HO cutting efficiency (S5C Fig). (E) The sensitivity of the *esa1-ts* mutant cells to DNA damage is partially suppressed by deletion of core NHEJ factors. 10-fold serial dilution spot assays were performed with *esa1-L254P*, *rad9Δyku80Δ* and *rad9Δyku80Δesa1-L254P* cells in the absence or the presence of DNA damaging agent MMS. (F) The *esa1-ts* mutation leads to a decrease of DSB repair by resection-based single-strand annealing. A yeast strain with direct repeats 20kb apart is used as reporter for SSA efficiency. In the absence of Rad51 the DSN induced by HO between the repeats is mostly repaired by SSA with Rad52. Efficiency of repair is monitored by survival with 10-fold dilution spot assays in conditions of *GAL-HO* induction or not.

in the *nej1Δ* background, albeit to a lower level similar to the y*ku80Δ* single mutant (**Fig 5A and 5B**). Similar to a previous study [48], we also observed recruitment of Rad51 in y*ku80Δ* and *rad9Δyku80Δ* G1-blocked cells (**S5A Fig**).

To determine whether NuA4 has a role in this G1 phase mechanism of DSB repair, we introduced the *esa1-L254P* temperature sensitive (ts) mutant allele which affects the acetylation capacity of NuA4 even at semi-restrictive temperature (as shown by western blot for the H4ac in **Fig 5C**)[10]. Interestingly, Rfa1 ChIP-qPCR to measure the hyper-resection phenotype in the triple *rad9Δyku80Δesa1-ts* triple mutant background showed a significant reduction of DNA end resection compared to the *rad9Δyku80Δ* double mutant (**Fig 5D**). These results indicate that NuA4 acetyltransferase activity is important to stimulate DNA end resection in G1, likely in part through acetylation of chromatin in front of the resection machinery. This was supported by a partial suppression of the DNA damage sensitivity of *esa1-ts* mutant cells by the *rad9Δyku80Δ* deletions (**Fig 5E**). On the other hand, we did not observe a clear effect of NuA4 partial loss of function on Rad52 recruitment in these conditions in G1 cells (**S5B Fig**).

To directly assess the possible role of NuA4 in the repair of DSBs by the SSA pathway, we used a yeast reporter strain carrying two regions of homology (direct repeats) separated by 20 kb [49]. When the Rad51 recombinase is removed from this background, repair of the HO break occurs by the SSA pathway that requires Rad52. Growth (spot assays) in conditions of HO induction is used as measure of DNA repair. Deletion of *RAD52* in this background created a strong growth defect after HO induction, as expected since SSA is abrogated (**Fig 5F**). Importantly, the *esa1 ts* allele in combination with *RAD51* deletion led to a similar growth defect upon HO induction, suggesting that NuA4 is important for repair of DNA breaks by the resection-dependent SSA pathway (**Fig 5F**).

## Acetylation of Nej1 by NuA4 as a mechanism to inhibit NHEJ, favoring resection-based repair

Yeast Nej1 and its mammalian homologs XLF/NHEJ1 and PAXX have been shown to be important for repair by NHEJ [50–53]. Nej1 acts as a scaffolding protein for other NHEJ factors, the N-terminal region of Nej1 interacting with Yku and its C-terminal region with Lif1 [54–56]. XLF has also been shown to interact with mammalian Ku and XRCC4 [54]. Both Nej1 and Dnl4-Lif1 are required for retention/stabilization of Yku at DSBs [48,57]. Nej1, like its mammalian homolog, is important for tethering the two broken ends of the DNA [58,59] and it was also recently shown to interact with Mre11 of the MRX complex for its tethering activity [60].

In a large-scale *in vitro* acetylation assay on microarrays, Nej1 was shown to be a target of NuA4 acetyltransferase activity [17]. To investigate this further, we directly tested a recombinant Nej1 protein in acetyltransferase assays *in vitro* with purified NuA4 complex. Migration of the radioactive assay on gel and fluorography clearly indicated efficient acetylation of Nej1 by NuA4 (**Fig 6A**). To determine if Nej1 is indeed acetylated on lysine residues in vivo, we performed immunoprecipitations with an anti-acetyl-lysine antibody on extracts from cells expressing or not a Nej1 fusion protein in normal growth conditions or after MMS-induced DNA damage. The results showed that a small proportion of Nej1 seems indeed acetylated *in vivo* while DNA damage has no impact on the level of modification (**Fig 6B**). We then purified this Nej1 fusion protein from yeast cells and compared the acetylated lysine-containing peptides identified by mass spectrometry from wild type and the *esa1-L254P* NuA4 mutant backgrounds (**S2 Table**). Three lysine residues, at positions K18, K192 and K234, were detected as being acetylated on Nej1 in WT conditions but not in the mutant background (**Figs 6C** and

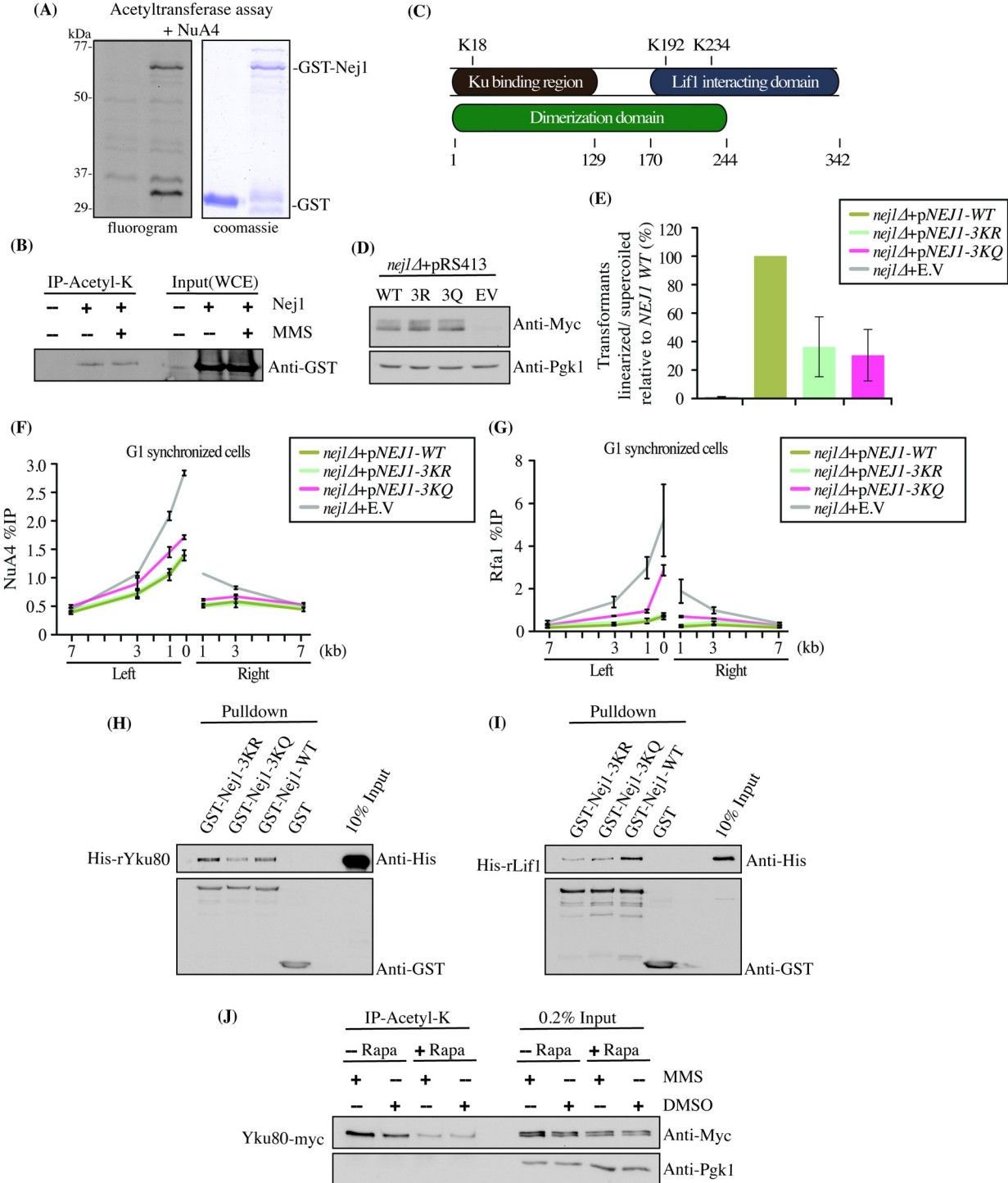

**Fig 6. NuA4 inhibits NHEJ through acetylation of Nej1.** (A) *in vitro* acetylation assay using purified NuA4 complex and recombinant GST-Nej1. After the reaction with radioactive acetyl-CoA, the reactions were loaded on gel, processed for fluorography/exposed on film. The GST-Nej1 protein is clearly acetylated by NuA4 *in vitro*.GST alone is used as negative control. Faint labeled bands in both samples correspond to known auto-acetylation of NuA4 subunits (e.g. Yng2 at 36kDa [17]). (B) Nej1 is acetylated *in vivo*. Anti-acetyl-lysine immunoprecipitations were performed on extracts from cells over-expressing GST-NEJ1 from a multicopy vector. Nej1 Acetylation level is detected by anti-GST western on the immunoprecipitated material. (C) Drawing showing the locations of the acetylated residues identified by mass spectrometry that were not present in the *esa1-ts* sample (see also S6A Fig and S2 Table for full list of identified acetylated peptides). Known functional domains of Nej1 are also indicated. (D) Western blot on whole cell extracts comparing native wild type and mutant Nej1 expression levels in constructed strains used in (E). (E) *nej1-3KR* and *nej1-3KQ* mutant cells show reduced plasmid re-ligation efficiency for EcoR1 linearized pSO98 plasmid. Samples were

normalized to wild type, which was set at 100%. (F-G) Increased NuA4 and resection/RPA recruitment in the *nej1-3KQ* mutant compared to wild type *NEJ1* and the *nej1-3KR* mutant background. ChIP-qPCR using anti-Eaf1 (F) and anti-Rfa1 (G) antibodies in G1 synchronized cells as described in Fig 1. Yeast strains used showed similar HO cutting efficiency (S6C Fig). (H) GST pull-down with recombinant proteins. Recombinant Nej1-3KQ shows reduced interaction with rYku80 compared to Nej1-WT and Nej1-3KR. rNej1 is GST tagged for pull-down whereas rKu80 is 6XHis tagged for detection by western. Empty GST was used as negative control. (I) GST pull-down as in (H). Recombinant Nej1-3KQ and Nej1-3KR show reduced interaction with rLif1 compared to wild type Nej1. rLif1 is 6XHis tagged. Empty GST was used as negative control. (J) Yku80 is acetylated on lysine residues *in vivo* in a NuA4-dependent manner. Acetyl-lysine immunoprecipitation in an Esa1-FRB anchor-away[24]/Yku80-13myc background shows much reduced signal for Yku80 after the rapamycin treatment that depletes Esa1 from the nucleus. The effect of DNA damage was also monitored by treating the cells with MMS. Pgk1 was used as loading control. The efficiency of Esa1 depletion was monitored by western blot showing loss of H4 acetylation (S6G Fig).

S6A). To determine if these acetylation sites are important for Nej1 function, we produced yeast strains that express physiological levels of Nej1 with non-acetylatable (K to R) or acetylation-mimic (K to Q) substitutions. Importantly, the point mutations did not affect the stability of Nej1 (Fig 6D). Next, we tested if these lysines affect the efficiency of DSB repair by NHEJ. As shown in Fig 6E, loss of Nej1 completely abrogated repair by NHEJ, measured by re-ligation efficiency of digested plasmid templates. Interestingly, both acetyl (3KQ) as well as non-acetyl (3KR) substitution mutants showed significantly decreased NHEJ efficiency. These results demonstrate the importance of these three lysine residues in Nej1 function during DSB repair by NHEJ. In contrast, Nej1 wild type and mutants show similar repair efficiency for a chromosomal DSB at the *MAT* locus (S6B Fig). Importantly, ChIP-qPCRs for NuA4 and RPA around the HO-induced DSB in G1 cells showed significantly increased recruitment of NuA4 as well as resection in the acetyl mimicking *nej1* mutant (Fig 6F and 6G; the non-acetyl *nej1* mutant datapoints overlap with the wild type ones). On the other hand, the increases seen in the *nej1-3KQ* background did not reach the levels obtained with the complete loss of Nej1. This could be due to Nej1-3KQ still getting recruited at the break, but with impaired function. Since Nej1 interaction with other NHEJ factors is important for its function [55–57,60,61], we performed *in vitro* protein-protein interaction analysis using Nej1 mutants with recombinants Yku80 and Lif1. While wild-type Nej1 and the 3KR mutant showed similar interaction with Yku80, the acetyl mimicking 3KQ mutant showed less binding (Fig 6H). In parallel, interaction of Nej1 with Lif1 was significantly decreased in both 3KQ and 3KR mutants compared to wild type (Fig 6I). The decreased binding of Nej1-3KR with Lif1 but not with Yku80 could explain its effect on the plasmid re-ligation assay (Fig 6E). Lif1 binds the Dnl4 ligase and loss of a stable interaction with Nej1 may result in defective Lif1-Dnl4 recruitment, leading to impaired DSB religation. However, since Nej1-3KR is still able to interact with Yku80, stabilizing it at the break, it can explain the absence of effect on resection compared to wild type cells (Fig 6G). Furthermore, the Nej1-3KQ acetyl-mimic mutant also showed decreased binding to the C-terminal part of Mre11 compared to wild type and the 3KR mutant, which may affect its tethering activity (S6D Fig). Importantly, Nej1 phosphorylation in response to DNA damaging agents [62] was not affected by the 3 lysine substitutions or the depletion of NuA4 acetyl-transferase activity (Esa1) using the anchor-away system [63,64](cells were treated with DNA damaging agent MMS, phosphorylation leading to the appearance of a slower migrating Nej1 signal) (S6E–S6G Fig).

In mammals, NHEJ core factor Ku70 is known to get acetylated by CBP which inhibits DNA repair and regulates Bax-mediated cell apoptosis [65,66]. We tested if yeast Ku is also acetylated in yeast cells and if it could be a non-histone target of NuA4, along with Nej1. Using the anchor-away system to deplete Esa1 (S6G Fig) and immunoprecipitation with an anti-acetyl-lysine antibody, we found that Yku80 was indeed acetylated *in vivo* and NuA4 was clearly required for most of this acetylation to occur (Fig 6J). Interestingly, Yku80 acetylation seemed slightly higher in conditions of DNA damage. Taken together, these data indicate that Nej1

and Yku80 are non-histone substrates of NuA4 acetyltransferase activity. Acetylation of Nej1 may reduce NHEJ efficiency by disrupting its interaction with Yku80 and Lif1. Further studies are required to characterize the effect of acetylation on Yku80 function but work in higher eukaryotes also supports an inhibitory role.

## Discussion

Altogether, the results of this study allowed us to clearly demonstrate a reciprocal antagonistic relationship between the NuA4 complex and the NHEJ machinery, confirming earlier genetic interaction data and our work with mammalian homologs. The *in vitro* and *in vivo* data also went much farther by showing a clear inhibition of NuA4 recruitment and activity on chromatin by Rad9 and NHEJ factors. Strikingly, In the other direction, our data showed that NuA4 can directly inhibits the NHEJ machinery by acetylating some of them. Through this work, we also uncovered a new alternate mode of recruitment for NuA4 at DSBs in specific conditions. The Xrs2-dependent pathway is responsible for initial recruitment of NuA4 in G1 (**Fig 2**), as we previously reported in S/G2 [23]. This happens early during DSB signaling and the repair process when MRX is recruited to the break site. It is known that PIKK family member Tra1, subunit of NuA4, can directly interact with Xrs2 [23]. The other main acetyltransferase complex in yeast, SAGA, also contains Tra1 and requires Xrs2 for its recruitment at the break [24]. Tra1 is therefore related to the Tel1(ATM) signaling kinase not only at the amino acid sequence but also in its mode of recruitment at the DNA break being dependent on Xrs2 [42]. Tel1 signals the DSB by phosphorylation of H2A-S129 ($\gamma$-H2A) in the surrounding chromatin [67]. The nucleolytic processing, termed resection of the DNA breaks produces 3' ssDNA ends. This ssDNA gets coated by RPA to prevent secondary structure formation and degradation. RPA-coated ssDNA is recognized by various repair proteins, one such being Lcd1/Ddc2 (ATRIP) [3,44]. We had shown previously that Tra1 can also directly interact with Lcd1 *in vitro* [23]. Our *in vivo* data now support this link as NuA4 can still be recruited at the break in specific conditions where Xrs2 function is genetically bypassed and resection can occur, and this recruitment requires Lcd1/Ddc2 (**Figs 3 and 4**). Since the simple deletion of *LCD1/DDC2* in normal conditions had no effect on NuA4 recruitment at DSBs [23](**S4B Fig**), this indicates that Lcd1/Ddc2 is an alternate mode of recruitment occurring only in specific conditions. Nevertheless, it is tempting to speculate a function of this mechanism in the spreading of NuA4 along resection on each side of the break after initial recruitment by MRX (the two-step recruitment model described in [23]). This alternate mode may also be more important specifically during G1 in hyper-resection conditions when Rad9 is absent and the NHEJ machinery is defective.

Resection is the event that marks the commitment of DSB repair to the HR process. The 3' ended ssDNA searches for the homologous sequence or sister chromatid to use as a template to complete the repair process [3]. Resection is a very regulated process and is active in S/G2 phase when the Cdk1 activity is high and inhibited during the G1 phase of the cell cycle when Cdk1 activity is low [37]. But it has been shown previously that resection can be achieved in G1 phase with the deletion of Ku protein [37,48]. In mammalian cells, HR in G1 phase requires deletion of the *RAD9* homolog 53BP1, depletion of KEAP1 and a phospho-mimicking mutant of CtIP [68]. In our experiments we detect efficient NuA4 recruitment at a DSB in G1 concomitant with the appearance of resection (**Fig 1**). It was previously shown that deletion of NHEJ factors in G1 not only leads to resection but also repair of the DSB by SSA, relying on the recombination protein Rad52 [37]. Our question then was to evaluate if NuA4 played a role in this resection-based repair in G1. Temperature sensitive mutant of Esa1, the catalytic subunit of NuA4, slows down resection which likely impairs such repair process requiring extended

end resection (**Fig 5D**). Indeed, using a SSA reporter strain, the *esa1* defective allele creates a clear growth defect when the HO break is induced in that background (**Fig 5F**). Importantly, this implicates the NuA4 complex in another DSB repair pathway beside HR. It will be interesting to verify if the SAGA HAT complex also collaborates with NuA4 in this pathway, like it does in HR [24].

The antagonistic relationship between NuA4 and the NHEJ machinery/Rad9 seems central to the mechanisms regulating the repair pathway choice. The fact that two NHEJ core factors, Nej1 and Yku80, are new non-histone targets of NuA4 acetyltransferase activity (**Fig 6**) is likely an important part of these regulatory events. Nej1 is critical for repair by NHEJ [50,51,53] and our *nej1* acetyl mutants also show impaired NHEJ (**Fig 6E**). Nej1 is known to act as a scaffold for binding and stabilizing other proteins of the NHEJ machinery [54–57]. Nej1 acetyl-mimicking mutant shows reduced binding to Lif1 and Yku80 *in vitro*, which correlates with increased resection seen *in vivo* (**Fig 6G**). It will be interesting to determine how these individual lysines affect interaction of Nej1 with other NHEJ proteins and whether the tethering activity of Nej1 itself is impaired in these mutants [60]. In both yeast and mammals, it has been documented that Ku may interact with various histone deacetylases (HDACs) and these have an important role in repair by NHEJ [69–71]. Further analysis of NuA4 interplay with NHEJ would include identification of the specific lysines on Yku80 that get acetylated by NuA4. Removal of positive charge of lysines through acetylation may reduce or abrogate the binding capacity of Yku80 to DNA at the break site leading to processing/ resection of the DNA ends. The acetylation may also affect the binding with Yku70 and formation of the Ku complex, thereby impairing NHEJ.

**Fig 7** presents a model, based on our data, for the antagonistic relationship between NuA4 and Rad9/NHEJ factors at DNA breaks. NuA4 can be recruited by the MRX complex in the early steps after DSB formation. Since Rad9 and NHEJ factors get assembled at the break, these, apart from inhibiting resection, may also inhibit recruitment of NuA4 and acetylation of chromatin. If NuA4 is able to directly acetylate Nej1, this blocks the assembly of the NHEJ repair machinery, thereby favoring the initiation of DNA end resection. Resection then leads to RPA-coated ssDNA, recruitment of Lcd1/Ddc2(ATRIP) and NuA4 spreading on each side away from the break, helping long-range resection in the context of chromatin. In the situation of Xrs2 absence or bypass, NuA4 can also be recruited by Lcd1/Ddc2. Our results clearly demonstrate that NuA4 regulates the repair pathway choice, away from NHEJ, and can favor repair by resection-based processes throughout the cell cycle. It supports and greatly broadens the conclusions of our previous work that highlighted a crucial battle between 53BP1 (mammalian Rad9) and NuA4/TIP60 to regulate DSB repair pathway choice. Future work will verify if the mammalian NuA4/TIP60 complex also targets NHEJ proteins during regulation of repair pathway choice. Interestingly, it has been recently shown that 53BP1 can also be acetylated *in vivo*, which inhibits repair by NHEJ and favors HR [72]. It is also possible that NuA4 is involved in other DSB repair pathways such as alt-NHEJ, underpinning its role in resection-based repair. A better understanding of how NuA4/TIP60 gets recruited to the DNA damage site and acetylates specific non-histone proteins to modulate repair, identifies new molecular targets that can be acted upon to treat diseases such as cancer.

## Methods

### Yeast strains and growth assay

Yeast strains used in the current study are listed in **S1 Table**. Standard PCR based protocol was used for respective transformations employing the lithium acetate method. Yeast cells were grown in YPD (1% yeast extract, 2% peptone, 2% dextrose) at 30°C or in synthetic

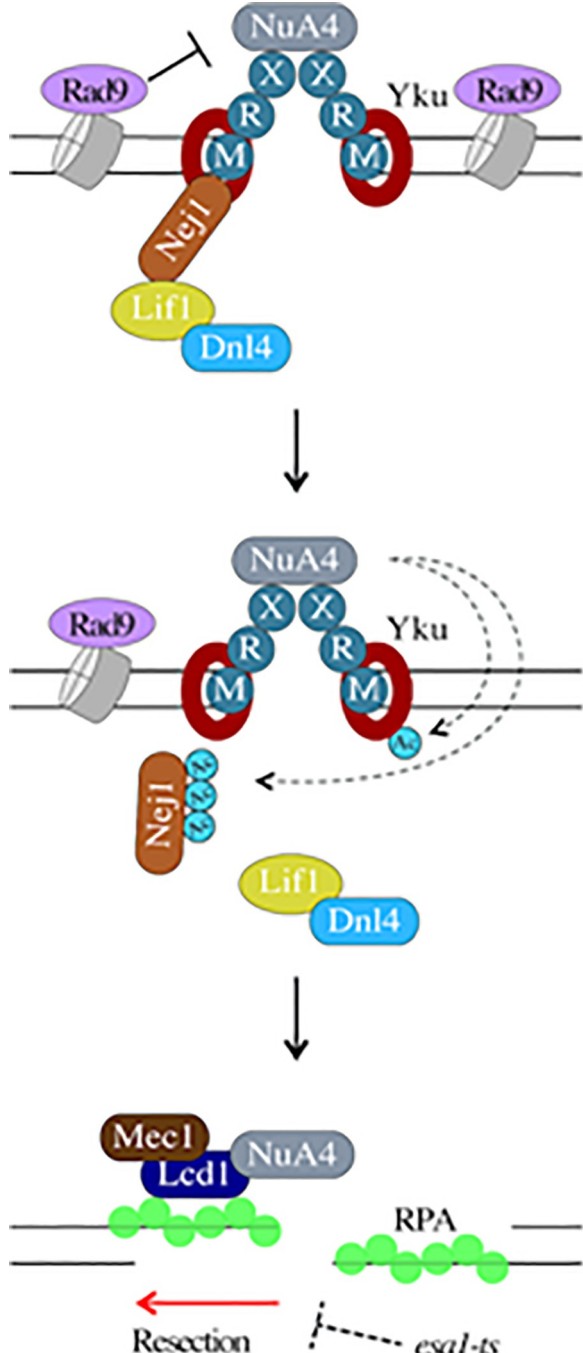

**Fig 7. Reciprocal antagonism between the NuA4 complex and the NHEJ machinery: a central regulatory mechanism for repair pathway choice.** Model: While chromatin-bound Rad9 can inhibit NuA4 recruitment and chromatin acetylation to favor repair by NHEJ, recruitment of NuA4 by MRX can lead to acetylation of NHEJ core factors like Nej1 and Yku80. This perturbs the interactions between NHEJ factors, inhibiting NHEJ and enabling resection of DNA ends. As resection produces ssDNA, NuA4 spreads on each side of the break, favoring resection-based repair pathways like HR and SSA. The possible alternate mode of NuA4 recruitment by Lcd1/Ddc2 in specific conditions is also depicted. "Ac" represents acetylation performed by NuA4.

complete media. For DSB induction, cells were grown at 30˚C overnight in YP-Raffinose and 2% galactose was added for 3 hours to induce HO endonuclease.

Yeast spot assays were performed with yeast culture grown overnight in YPD at 23˚C and SC-URA at 30˚C, inoculated in the morning in YPD/ SC-URA at $OD_{600}$ 0.3 and grown for 3–4 hours. The 10-fold serial dilutions of the yeast cultures were made for spotting on YPD/ SC-URA plates supplemented with methylmethane sulfonate (MMS). Colonies were allowed to grow for 3–4 days at 32˚C for YPD and 30˚C for SC-URA plates and the growth sensitivities were compared to YPD/ SC-URA control plate.

## Chromatin Immunoprecipitation (ChIP)

ChIP-qPCR was performed as described previously [15]. In short, yeast cells were grown in YP-Raff till optical density (O.D. 600) of the culture was between ~0.5 to 1. α-factor was added to the indicated experiments for 6 hours to synchronize the cells in G1 phase. This was followed by addition to galactose for 3 hours to induce DSB at MAT locus. Next, cells were cross-linked with formaldehyde and cells were lysed by bead beating. Sonication was performed (Diagenode Bioruptor) to obtain chromatin size of ~200–500 bp. 100ug of chromatin was incubated with the indicated antibody overnight on a wheel at 4˚C. Next day, protein A or G sepharose beads were added followed by subsequent washing steps and overnight at 65˚C for de-crosslinking. Phenol chloroform extraction was performed, and DNA samples were dissolved in NTE to be used for qPCR. LC480 LightCycler (Roche) was used to quantify DNA with the primer pairs flanking the indicated region left (0.15bp, 1Kb, 3Kb, 7Kb) and right (1kb, 3Kb, 7Kb) of induced DSB at MAT locus. Homothallic endonuclease (HO) cutting efficiency at MAT locus was calculated using qPCR with primer pair spanning the HO cut site normalized to uncut control site, interV (long intergenic region on chromosome V). A similar qPCR approach normalized to a control site is also used to directly measure resection on each side of the HO break, progressive loss of one strand decreasing by up to 2-fold the amount of template at different distance from the break. Primer sequences are available upon request. Antibodies used for ChIP are as follows: anti-Eaf1 [7], anti-Rfa1 (Agrisera AS 07214), anti-Rad51(Santa Cruz y-180), anti-Rad52(Santa Cruz yC-17).

## Recombinant protein purification and pull-down assay

Recombinant proteins were purified from BL-21 bacteria using standard protocol described previously [73]. In brief, bacterial culture was growth overnight at 16˚C in presence of IPTG. Bacterial pellet was treated with lysozyme and sonicated. Soluble fraction was incubated with Ni-nitrilotriacetic acid (NTA)-agarose (Qiagen) beads or glutathione-sepharose (GE Healthcare) for 3–4 hours on wheel at 4˚C for His- or GST-tagged proteins respectively. Coomassie blue staining was performed to estimate purified protein concentration with known bovine serum albumin (BSA) standards. After elution ~300–600 ng of His-tagged protein were pre-cleared with glutathione sepharose beads and incubated with immobilized GST-tagged protein in binding buffer (25 mM HEPES [pH 7.5], 100 mM NaCl, 10% glycerol, 100 μg/ml BSA, 1 mM PMSF, 0.5 mM dithiothreitol [DTT], 0.1% Tween 20, and protease inhibitors) at 4˚C for 3 hours followed by washing the beads three times. 1 μg of GST only immobilized protein was used as control. After washing Laemmli buffer 1X was added followed by boiling and loading on SDS-PAGE for visualization by western blotting using anti-His (Clonetech 631212) and anti-GST (Sigma G1160) antibodies.

## Purification of acetylated Nej1 for mass spectrometry

GST-tagged Nej1 was purified from wild-type and *esa1-L254P* (ts) background, followed by mass spectrometry analysis to identify the acetylation sites as described previously with few

modifications [17]. In short, cells were grown in SC-Ura containing 2% raffinose till $OD_{600}$ ~0.8–1.0. Galactose (2% final) was added for the induction of the GST-tagged Nej1 (expressed from a high-copy 2 micron vector) and cells were grown at 37˚C for two additional hours. Cells were harvested and lysed in extraction buffer (10mM Tris-HCl pH 8.0, 350mM NaCl, 0.1% NP-40, 10% glycerol, 1mM DTT, 1mM PMSF, 2 μg/mL leupeptin, 2 μg/mL pepstatin A, 5 μg/mL aprotinin, 5 mM β-glycerophosphate, 10 mM sodium butyrate, 5 mM NaF, 10mM nicotinamide, 15μM TSA). The lysate was centrifuged at 10,000g for 30min at 4˚C, followed by ultracentrifugation for 1 hour at 45,000g. The supernatant was precleared with CL6B beads (Sigma) before the addition of glutathione-sepharose beads (GE Healthcare) for 3 hours on wheel at 4˚C. After binding the glutathione-sepharose beads were washed 5 times with wash buffer (20mM Hepes pH7.5, 200mM NaCl, 10mM sodium butyrate). Beads were mixed with elution buffer (50mM Tris-HCl pH8.0, 10mM glutathione, 10% glycerol, 5mM sodium butyrate) at 4˚C on wheel and first elution was collected after 1 hour and other after overnight incubation. Next, protein A-sepharose beads were washed with IP lysis buffer (75mM Tris-HCl ph 7.5, 50mM NaCl, 1mM EDTA, 1mM EGTA, 10% glycerol, 1% Tween-20, 1mM DTT, 1mM PMSF, 5mM NaF, 2 μg/mL leupeptin, 2 μg/mL pepstatin A, 5 μg/mL aprotinin, 10mM sodium butyrate, 10mM, nicotinamide, 15μM TSA, 1 μM MG132) and conjugated with 6μl of mouse monoclonal anti-acetylated-lysine (Cell Signalling 9681S). Antibody conjugated protein A-sepharose beads were incubated with GST elution at 4˚C for 3 hours on a wheel. After washing 4 times with IP wash buffer (50mM Hepes pH 7.0, 100mM NaCl, 10 mM sodium butyrate) 20 μl of SDS-PAGE sample buffer was added and boiled. The samples were resolved by SDS-PAGE followed by colloidal Coomassie blue staining to visualize and cut gel bands, which were send for mass spectrometry analysis after trypsin digestion (Y. Zhao lab, Univ. of Chicago). Peptide coverage obtained was 77% for the WT sample and 83% for the mutant. List of confidently identified peptides containing an acetylated lysine residue is presented in S2 Table.

## Chromatin binding and acetyltransferase assays

Histone acetyltransferase assay was performed as described [9,74]. 500ng of short oligonucleosomes purified from methyl methanesulfonate MMS (0.05%) treated yeast cells, was incubated with recombinant Rad9 at 30˚C for 30 min. This was followed by the addition of purified NuA4 complex (23) and 0.125 μCi of [$^3$H]acetyl coenzyme A ([3H]acetyl-CoA) in HAT buffer (50 mM Tris-HCl [pH 8], 50 mM KCl plus NaCl, 0.1 mM EDTA, 5% glycerol, 1 mM DTT, 1 mM PMSF and 20 mM sodium butyrate) at 30˚C for 30 min. The reaction mixture was spotted on p81 filter paper, washed and air dried. The incorporation of radiolabeled acetylation was measured by scintillation counting. For in-gel assay with recombinant GST-Nej1, reactions with NuA4 were stopped in 1X Laemmli buffer, boiled and loaded on a 10% SDS-PAGE. Gel was treated with En3hance (Perkin Elmer), dried and exposed on film.

## Non-homologous end joining (NHEJ) assay

Plasmid re-ligation assay was performed as described [50]. 0.4 μg plasmid pSO98 supercoiled or EcoR1 digested was used to transform *nej1Δ* yeast strains that harbored wild type or mutant form of the *NEJ1* gene on a CEN:ARS low copy number plasmid. The transformed yeast was plated on SC-His-Leu and SC-His-Ura and kept at 30˚C. After 3 days colonies were counted and ratio of colonies transformed with digested plasmid to undigested supercoiled plasmid was taken. Data represents two independent biological replicates and error bar denotes the range between the biological replicates.

Chromosomal NHEJ assay was performed as described [51]. Yeast background used carries deletion of *HML* and *HMR* and galactose-inducible HO endonuclease that creates a single DSB at the *MAT* locus. Endogenous *NEJ1* gene was deleted and wild type or mutant form of the *NEJ1* gene was expressed from a low copy number CEN:ARS plasmid. The yeast cells were grown overnight in YP-Raffinose media. Next day appropriate number of cells in the log phase were plated on YP-Galactose and YP-Glucose plates and kept at 30˚C. After 3–4 days the number of colonies was quantified and ratio of colonies on YP-Galactose to YP-Glucose was calculated. Data are from three independent biological replicates and error bar denotes the standard deviation between the replicates.

## Acetyl-lysine immunoprecipitation

Immunoprecipitation was performed as described previously [23]. Cells were grown in YPD till OD600 was ~0.5 followed by rapamycin addition (1µg/ml final) for 30 mins. MMS (0.05%) or DMSO was added for 2 hours. Cells were collected and lysed in 10 mM Tris-HCl pH 8.0, 150 mM NaCl, 10% glycerol, 0.1% NP-40, 2 µg/mL leupeptin, 2 µg/mL pepstatin A, 5 µg/mL aprotinin, 1 mM PMSF, 10 mM β-glycerophosphate, 1 mM Sodium Butyrate, 0.5 mM NaF, and 1 mM DTT. 4 µg of WCE was incubated with anti-acetyl (ImmuneChem ICP0380), overnight at 4˚C. Protein G Dynabeads (Invitrogen 10004D) was added to the reaction and kept on wheel at 4˚C for 4 additional hours. The IPs were washed and 1X Laemmli buffer was added and boiled. The samples were resolved on SDS-PAGE followed by western blotting and analyzed with anti-Myc (9E10 Babco MMS150R) and anti-Pgk1 (Abcam 113687).

## Supporting information

**S1 Fig. Inhibition of NuA4 by Rad9 in vitro but no inhibition of NuA4 and resection spreading by NHEJ factors in asynchronous cells.** Related to Fig 1. (A) Purified recombinant Rad9 wild-type and mutants were resolved by SDS-PAGE and stained with Coomassie blue stain. Known concentrations of BSA were used to estimate the concentration of purified recombinant proteins using the ImageJ software. (B) Inhibition of NuA4-dependent acetylation of chromatin by Rad9 in vitro. The assays was performed as in Fig 1B but using more recombinant rad9 proteins. In these conditions the difference between WT and mutant Rad9 is more subtle because of mass effect, but still visible with 2ug. (C) ChIP-qPCR showing NuA4 enrichment around HO DSB at the *MAT* locus in wild type and *rad9Δ* in asynchronous cells. (D) ChIP-qPCR with antibody against Rfa1 showing no difference in resection at the *MAT* locus between wild type and *nej1Δ* in asynchronous cells. (E-I) HO cutting efficiency at the *MAT* locus after 3 hours of galactose induction in wild type and, *rad9Δ* in asynchronous cells (E), *rad9Δ* in G1 synchronized cells (F), *yku80Δ* and *yku80Δrad9Δ* in G1 synchronized cells (G), *nej1Δ* in G1 synchronized cells (H), *nej1Δ* in asynchronous cells (I).
(TIF)

**S2 Fig. Difference in NuA4 and resection signal seen between wild type and mutants not caused by uneven HO cutting at *MAT* locus and a different method to measure DNA end resection in parallel to ChIP-qPCR with Rfa1.** (A-C) HO cutting efficiency at MAT locus in G1 synchronized cells after 3 hours of galactose induction between wild type and, *xrs2Δrad9Δ* (A), *xrs2Δrad9Δyku80Δ* (B), *mre11-H125N* (C). (D) DNA end resection is directly measured by decreased DNA signal near the break in WT, *xrs2Δ*, *yku80Δrad9Δ* and *xrs2Δrad9Δyku80Δ* backgrounds. Hyper resection phenotype is clearly seen with *yku80/rad9* double mutant but disappear in the absence of Xrs2. Resection is presented as % of Intact DNA determined by qPCR on genomic DNA at indicated sites and normalized to the negative control intergenic V.

Cells were grown in YP-Raff till early log phase followed by addition of galactose for 3 hours to induce the DSB.
(TIF)

**S3 Fig. p*MRE11-NLS* suppresses defect of *xrs2*Δ.** (A) Phenotypic analysis showing introduction of p*MRE11-NLS* suppresses the sensitivity of *xrs2*Δ cells to the DNA damaging drug MMS. 10-fold serial dilutions of log phase cells were spotted on synthetic complete -URA plates with and without MMS. (B-D) HO cutting efficiency at *MAT* locus after 3 hours of galactose induction in the indicated strains.
(TIF)

**S4 Fig. Lcd1/Ddc2 is not required for NuA4 recruitment or resection in asynchronous cells.** (A-B) ChIP-qPCR showing that the *lcd1*Δ *xrs2*Δ background results in loss of RPA signal (A), and NuA4 (B) at the HO induced DSB in asynchronous cells. The single mutant *lcd1*Δ (loss of Mec1) does not affect resection but shows slightly less NuA4 signal compared to wild type close to the break. (C-D) HO cutting efficiency at *MAT* locus after 3 hours of galactose induction in the indicated strains.
(TIF)

**S5 Fig. The *esa1-ts* allele does not affect the recruitment of recombination protein Rad52 in a hyper resection background in G1.** (A) ChIP-qPCR with a Rad51 antibody around the induced DSB at *MAT* locus in G1 synchronized cells. y*ku80*Δ and y*ku80*Δ*rad9*Δ cells show higher signal compared to wild type, linked to hyper-resection. (B) ChIP-qPCR with a Rad52 antibody around the induced DSB at *MAT* locus in G1 synchronized cells. *rad9Δyku80Δ* and *rad9Δyku80Δesa1-ts* cells show higher signal for Rad52 than wild type but no significant difference by the addition of the *esa1 ts* allele. (C) HO cutting efficiency in wild type, *rad9Δyku80Δ* and *rad9Δyku80Δesa1-ts* cells at the *MAT* locus in G1 synchronized cells after 3 hours of galactose induction.
(TIF)

**S6 Fig. Nej1 acetylation mutants do not disrupt DNA damage induced phosphorylation but affect binding to Mre11.** (A) Nej1 is acetylated *in vivo* on residues K18, K192 and K234 in a NuA4-dependent manner. Mass spectrometry analysis of Nej1 lysine acetylation sites in *ESA1* wild type and temperature sensitive (ts) mutant. See full peptide sequences in S2 Table. (B) Nej1 wild-type and lysine mutants show similar ability to repair a chromosomal DSB induced at *MAT* locus by HO induction in the absence of *HML/HMR* donor sequence. (C) HO cutting efficiency at *MAT* locus in G1 synchronized cells after 3 hours of galactose induction in the indicated *nej1*Δ strains. (D) *In vitro* protein pull-down assay showing that recombinant wild type Nej1 and Nej1-3KR bind to the C-terminal region of rMre11, while this interaction is much reduced with Nej1-3KQ. Empty GST was used as negative control. (E) Nej1 lysine mutants do not affect the DNA damage induced phosphorylation of Nej1. Western blot analysis of whole cell extracts (WCE) prepared from indicated yeast strains treated with MMS (0.05%) or DMSO (control) for 2 hours. Pgk1 was used as loading control. (F) Western blot analysis on WCE showing depletion of Esa1 does not affect the DNA damage induced phosphorylation of Nej1. Esa1 was FRB tagged and depleted from the nucleus using the anchor-away system in presence of rapamycin [24]. This was followed by treatment of cells with MMS or DMSO. Pgk1 was used as loading control. (G) Western blot analysis of yeast WCE showing reduced acetylation of H4 in the anchor-away background with Esa1 FRB tagged after rapamycin treatment. H2B was used as loading control.
(TIF)

**S1 Table. Yeast strains used in the current study.**
(DOCX)

**S2 Table. Acetylated peptides detected by mass spectrometry in *ESA1* and *esa1* ts strains.**
List of Nej1 peptides containing an acetylated lysine residue confidently identified by mass spectrometry after affinity purification from extracts of WT and *esa1 ts* cells and trypsin digestion. Total peptide sequence coverage obtained were 77 and 83% respectively.
(DOCX)

## Acknowledgments

We thank Olivier Jobin-Robitaille pour early work on this project, Zhiyou Deng and Yingming Zhao (U. of Chicago) for mass spectrometry analysis of purified GST-Nej1. We thank James E. Haber and Jennifer Cobbs for stimulating discussions. We are grateful to James E. Haber and Lorraine S. Symington for providing strains and plasmids.

## Author Contributions

**Conceptualization:** Salar Ahmad, Jacques Côté.

**Formal analysis:** Salar Ahmad, Jacques Côté.

**Funding acquisition:** Jacques Côté.

**Investigation:** Salar Ahmad, Valérie Côté, Xue Cheng, Gaëlle Bourriquen, Vasileia Sapountzi, Mohammed Altaf.

**Methodology:** Salar Ahmad, Xue Cheng.

**Project administration:** Jacques Côté.

**Supervision:** Jacques Côté.

**Writing – original draft:** Salar Ahmad, Jacques Côté.

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
