## [Decision Letter · Decision Letter 0]

17 Mar 2021

Dear Dr Côté,

Thank you very much for submitting your Research Article entitled 'Antagonistic relationship of NuA4 with the Non-Homologous End-Joining machinery at DNA damage sites' to PLOS Genetics.

The manuscript was fully evaluated at the editorial level and by independent peer reviewers. The reviewers appreciated the attention to an important problem, but raised some substantial concerns about the current manuscript. Based on the reviews, we will not be able to accept this version of the manuscript, but we would be willing to review a much-revised version. We cannot, of course, promise publication at that time.

If you decide to revise the manuscript for further consideration at PLOS Genetics, please aim to resubmit within the next 60 days, unless it will take extra time to address the concerns of the reviewers, in which case we would appreciate an expected resubmission date by email to plosgenetics@plos.org.

[LINK]

We are sorry that we cannot be more positive about your manuscript at this stage. Please do not hesitate to contact us if you have any concerns or questions.

Yours sincerely,

Lorraine S. Symington

Associate Editor

PLOS Genetics

Gregory P. Copenhaver

Editor-in-Chief

PLOS Genetics

Reviewer's Responses to Questions

**Comments to the Authors:**

Reviewer #1: In this paper, Ahmad et al. study DSB processing and NuA4 binding in the G1 phase. They use rad9 and ku mutants to achieve conditions of decreased end protection and increased resection and show that increased resection goes hand-in-hand with increased NuA4 binding. Interestingly, they show that an esa1 acetylation defective mutant reduces the hyper-resection of a rad9 (or rad9 ku80) mutant and conclude that NuA4 acetyltransferase activity is important to stimulate DNA end resection in G1. Presumably this same role would operate in S/G2 or other situations where resection occurs at a DSB, but this wasn’t addressed. They characterize the resection requirements thoroughly, and establish that recruitment of NuA4 depends on Xrs2, but ultimately this may be due to an indirect effect of lack of Mre11 entry to the nucleus and resection in the Xrs2 mutant, and maybe Ddc2 is the actual NuA4 interactor. The Ddc2 involvement data is indirect and the role of Xrs2 vs Ddc2 is not clearly distinguished, either experimentally or in the writing. Finally, they establish that end-joining factors Nej1 and Yku80 are NuA4 (Esa1) targets, and acetylation of Nej1 reduces its interaction with other NHEJ factors. This could provide a way to control DSB repair pathway choice. This part of the paper was under-developed and not connected mechanistically to the rest.

Overall, this paper describes 2 somewhat separate roles for NuA4 in NHEJ, neither of which was fully characterized. (1) (Fig. 5) direct acetylation of members of the Ku end-joining complex to regulate their interactions (and possibly stability on the DNA, though this was not directly shown). (2) Figs 1-4. Role of NuA4 in resection. This is presumably not a normal G1 role but can occur in G1 in abnormal circumstances such as when end protection is gone (ku80 missing) or resection is uncontrolled (rad9 deletion). However, it was not clearly established how NuA4 is recruited (indirect evidence that it could be through Ddc2 interaction) or what it is acetylating to increase resection (what are the targets). Since this was done in conditions where the end was already unprotected, presumably the targets are RPA or histones or some other internally binding protein. Thus, this part of the story seems un-connected to the role of acetylating the Ku/end-joining factors. The authors could consider separating these data into two manuscripts and more thoroughly characterizing each pathway.

Fig. 1B. Is the difference between wt and mutant rad9 conditions statistically significant?

Fig. 2E. The Mre11-H125N data shows that NuA4 is still recruited when resection is less robust (resection is not abolished in this mutant). However, I don’t see how this shows that Xrs2 is the protein recruiting NuA4. And the data in Fig. 3 indicate that the main role of Xrs2 re. NuA4 recruitment is to bring Mre11 into the nucleus to start the resection process. Therefore, the conclusion of an Xrs2 direct role is not substantiated here. However, previous data showed a direct interaction between Xrs2 and NuA4 subunit Tra1 and concluded there was direct recruitment by Xrs2. This discrepancy is not directly addressed in the text, leading the reader confused about what the authors believe is the role for Xrs2.

Fig. 3 text. I think the authors are implicating Lcd1/Ddc2 as the actual NuA4 recruiter, but it is not clearly stated how the absence of Mre11/Xrs2 would lead to insufficient Ddc2 recruitment. I assume it is indirect, that you need Mre11-dependent resection and RPA loading to recruit Ddc2, which in turn recruits NuA4. This could be more clearly written in the text. (side note: isn’t Ddc2 the more commonly used name?)

Fig. 3E-F. A NuA4 and RPA ChIP in the lcd12sml1 double should be done to show that Ddc2/Lcd1 is responsible for NuA4 recruitment. If there isn’t enough resection in G1 cells, then ku80 and/or rad9 can be also deleted. This seems much more straightforward than the experiment done, which doesn’t show very clear results. The “increased resection” seen when adding in the pEXO1 plasmid is very subtle. And the experiment in 3F doesn’t have a Lcd1+ condition to compare to, which needs to be included. Altogether, the data supporting recruitment of NuA4 by Lcd1/Ddc2 is quite indirect. Since this is a major point of the paper, this conclusion should be better supported. Confusingly, when I looked back at the data from the previous paper (ref 23), it turns out it was already shown that deletion of LCD1 had no effect on NuA4 recruitment to an HO break using a similar ChIP method. So why are the authors now speculating that Lcd1 binding is a second pathway for NuA4 recruitment to breaks?

To further support that an in vivo role of Ddc2 is to recruit NuA4, it would be helpful to delete an interaction domain on one of the proteins. For example, mutation of the Tra1 PIKK domain or the interacting Ddc2 domain to show that NuA4 recruitment to a DSB in is abolished. This could also get around any potential poor growth issues in using lcd1sml1rad9 triple mutants in the above suggestion (though lcd1sml1ku80 should not be sick).

It would be very interesting to combine the Xrs2-phospho interaction site implicated previously (ref 23) in NuA4 recruitment with a Ddc2 mutant to see if the two recruitment pathways work together or independently. Or does the Xrs2 phosphorylation allow Ddc2 recruitment, and so the former finding was also indirectly related to Ddc2 recruitment? Does NuA4 acetylation of RPA need to be included in the model?

Even if the role of Xrs2 vs Ddc2 in NuA4 recruitment is not addressed experimentally, it should be more clearly addressed in the writing. The introduction says that Xrs2 was previously implicated in NuA4 recruitment, and this was the premise of the experiment. But in the end, it seems that effect may be indirect (e.g no Xrs2, no Mre11 import to nucleus, no resection, so no RPA and Ddc2 loading to recruit NuA4). Altogether, the role of Xrs2/Mre11 versus Ddc2 needs to be clarified, taking into account both previously published and current data.

Fig. 5B is an experiment to show that Nej1 acetylation is important for NHEJ. The plasmid data is clear, but it would be important to check the phenotype in a chromosomal NHEJ assay. Similarly, it would be important to show a functional consequence on NHEJ upon the Esa1 depletion used in Fig. 5G that results in decreased Ku80 acetylation. These data are important, since a main point of the paper is that NuA4 acetylation of NHEJ factors regulates NHEJ. The title of this section “Acetylation of Nej1 by NuA4 in the regulation of DSB repair pathway choice” is misleading though, since a regulation of pathway choice is not shown.

Reviewer #2: see attachment

Reviewer #3: Ahmad et al. present an interesting study addressing the functions of the histone acetyltransferase NuA4 at DNA double strand breaks (DSBs), a relatively less studied aspect of DSB repair regulation. It builds from the group’s prior findings that suggest a competition between Rad9 (53BP1) and NuA4 (Tip60) at DSBs, which sets up the idea the NuA4 promotes resection, including that it is recruited by virtue of its Tel1-like subunit Tra1 in yeast interactions with Xrs2. Here they extend their findings by examining yeast G1 cells and interactions with the predominant G1 DSB repair pathway, nonhomologous end joining (NHEJ).

Results are novel and derived from experiments that are mostly well performed and yields new insights that Rad9 as well as Ku inhibit NuA4 binding in G1. Evidence is presented that both Lcd1/Ddc2 and Xrs2 collaborate to recruit NuA4 to DSBs. NuA4 activity is at least partially responsible for the resulting increased resection in G1, although definitive studies are impaired by the essential nature of NuA4 for cell viability. Novel acetylation targets for NuA4 are nominated as the NHEJ proteins Nej1 and Yku80. These data support a model in which NuA4 and NHEJ proteins, not unlike Rad9 and NuA4, have an antagonistic relationship in regulating DSB repair.

Overall, this is a quality study with interesting and novel findings that recommend it for publication. However, some significant issues should be addressed prior to publication.

1) In Fig. 2A, I assume the authors did not show rad9 mutant alone because it has already been shown previously? I think it is important and relevant to be able to see the quartet of 4 relevant mutants (WT, two singles, and one double) all on one plot. Similar comments apply to other plots where the reader is asked to compare traces between figures to get a full understanding of the strain relationships, which can be difficult to do accurately as presented.

2) Fig. 3 is a weaker part of the manuscript and difficult to interpret. The impact of the pMRE11-NLS plasmid clearly has a strong impact, but like my other comments I am having difficulty understanding if it provides a complete suppression of the xrs2 mutant phenotype regarding NuA4 recruitment. As above, this is because some allele combinations do not seem to be reported or are not on the same plot. Specifically, I am interested if xrs2 mutation still has any effect at all in strain carrying pMRE11-NLS? These results are important for the final interpretation that _both_ Xrs2 and Lcd1/Ddc2 independently recruit NuA4.

3) Continuing with Fig. 3, the strains in Fig. 3E,F having an accumulation of 5 different genetic alterations making it very difficult to understand and to be confident what the key interpretations are (even with the supporting text). Moreover, the key experimental manipulation is again missing, specifically LCD1 WT vs. lcd1 mutant with all of the other alleles in Fig. 3E,F still present. The authors seem to wish us to compare to other LCD1 WT strains, but those are not in the same complicated background. Only a side-by-side comparison with and without LCD1 can show its specific effect.

4) The main paper has only 6 Figures and one is a model only, which makes me wonder if some of the results in Supplemental ought to be promoted to the main paper. Fig. S3E may help address some of my concerns related to Fig. 3 above, in having more allele combinations, although I am still not certain all of the precise +/- Lcd1 and +/- Xrs2 combinations can be found.

5) If Fig 4., I would be interested to know if a specific SSA joint could be detected as forming within arrested G1 cells; since Rad52 is recruited, can productive repair be completed in G1 by a non-NHEJ mechanism? This is not a major point of concern, however.

6) Unless I missed finding it in the provided review materials, it is difficult or impossible to examine the evidence supporting the claim that the 3 lysine residues in Nej1 are in fact acetylated and to what extent in yeast cells. There is simply a table of identified sites in Supplemental. The mutant data are interesting but paint a picture of a pleiotropic effect that is not entirely attributable to acetylation (e.g. the impact of the K to R mutations on Lif1 binding), so it is important to be able to judge the quality of the evidence for acetylation independently of mutant effects that might be mediated by folding or other mechanisms. Might the anti-acetyl-lysine antibody be of use here, e.g. applied to WT and mutant Nej1 as was done for Ku?

7) No description or illustration is presented of where in the Nej1 protein the putative acetylated residues lie with regard to functional domains or three-dimensional structure. Such information would be a helpful adjunct to understanding the possible mechanistic consequence of any acetylation of those residues.

8) The authors use Rfa1 ChIP throughout as a surrogate for resection. This is reasonable for a study that uses ChIP extensively, but in one or two key conditions/comparison I would welcome DNA-based resection experiments in cells held in G1. Fig. 3E does something along these lines but the results are not compelling and unless I missed it the method used there is never described.

9) Throughout, the figures with line diagrams of signal at distances from the DSB use thin lines and tiny points that can be difficult to discern. At a minimum, thicker lines would make colors easier to discern, but I also recommend the use of visible data points with symbols that allow another way of telling traces apart.

10) The systematic gene names for the yeast Ku subunits have been YKU70 and YKU80 for many years now; I do not recommend even including the names HDF1 and HDF1 at this point.

11) Stylistically, I strongly recommend that the authors use proper scientific convention in always referring to results of the current study in past tense (i.e, “Deletion of XYZ1 resulted in”, not “results in”, “The effect was similar when”, not “is”, etc.).

12) There are grammatical errors throughout the paper, some of which I list below (there are likely more).

Page 5, .line 86, “antagonize”, not “antagonizes”.

Page 5, line 92, “allowing”, not “along”.

Page 5, line 96, “disrupts”, not “disrupt”

Page 8, line 157, “measured _by_ Rfa1/RPA”.

Page 8, line 162, “slow down _the_ rate”

Page 11, line 221, “terminus”, not “terminal”

Page 16, line 321, “thereby”, not “there by”

**Have all data underlying the figures and results presented in the manuscript been provided?**

Reviewer #1: **No: **ChIP data and other graphed data is only presented as an average +/- error on graphs. The underlying data is not in presented in numerical form, either averaged or individual experiments.

Reviewer #2: Yes

Reviewer #3: **No: **I was unable to find any detailed data supporting the discovery of Nej1 acetylated residues by mass spec, just a summary table of identified sites. See my review for additional comments; I find it ~impossible to judge the evidence that the sites are truly acetylated.

PLOS authors have the option to publish the peer review history of their article (what does this mean?). If published, this will include your full peer review and any attached files.

Reviewer #1: No

Reviewer #2: No

Reviewer #3: No

---

## [Decision Letter · Decision Letter 1]

23 Aug 2021

Dear Dr Côté,

Thank you very much for submitting your Research Article entitled 'Antagonistic relationship of NuA4 with the Non-Homologous End-Joining machinery at DNA damage sites' to PLOS Genetics.

The manuscript was fully evaluated at the editorial level and by independent peer reviewers. Two of the reviewers are satisfied by the revision but one reviewer expressed some concern that the conclusions drawn using G1 cells lacking anti-resection factors are overstated. We therefore ask you to modify the manuscript  address the specific points made by reviewer 1.

[LINK]

Yours sincerely,

Lorraine S. Symington

Associate Editor

PLOS Genetics

Gregory P. Copenhaver

Editor-in-Chief

PLOS Genetics

Reviewer's Responses to Questions

**Comments to the Authors:**

Reviewer #1: The paper has improved, but I still think that some conclusions are overstated and should be modulated.

(1) Concluding that NuA4 has a G1 role. The ChIP data was all collected in an un-natural condition, where G1 end protection is abolished and thus excessive resection is occurring, is misleading. For example, the conclusion at the end of the abstract “these results…suggests a role of NuA4 in alternative repair mechanism that involves DNA-end resection in G1”. The NuA4 role in G1 is only shown in the context of a mutant situation with increased DSB resection. The authors should stick to the conclusion that NuA4 acetylation antagonizes NHEJ through direct acetylation of NHEJ factors (Nej1 shown here). And that in cases of where DSB ends are available, be that in G1 due to a mutant situation, or maybe stochastically at some ends, or S/G2, NuA4 is recruited via interaction with Ddc2 and facilitates resection.

(2) Throughout the manuscript, the authors conclude that Xrs2 is required for NuA4 recruitment. Though recruitment is technically “dependent on Xrs2”, The data show that the Xrs2 results are mainly due to its role in Mre11 import and Mre11-dependent resection. It would be more informative to state that NuA4 recruitment is “dependent on MRX-mediated resection” or “dependent on Xrs2’s role in enabling MRX-mediated resection”. Key lines where this more accurate wording should be used are in the abstract line 31 and intro line 93.

Other writing issues:

Rad9 should not be characterized as a NHEJ factor (abstract and other places). But the point that Rad9 binding to resected DNA can directly inhibit NuA4 acetylation is interesting and can be made.

Reviewer #2: The authors have performed additional experiments and responded to reviewer's comments appropriately. They have revised manuscript and explain their work and model more clearly.

The experiments with pMRE11-NLS-X85 are a nice addition.

Reviewer #3: Ahmad et al present a revised version of their manuscript exploring the recruitment and activation of the NuA4 acetylase at DNB double strand breaks in G1, with a major conclusion that NuA4 and NHEJ promoting factors show an antagonistic relationship in which NuA4 acts to promote resection and is held in check by Rad9 and NHEJ proteins. In turn, NuA4 acetylates NHEJ proteins.

I was previously positive about the manuscript but had noted several problems with the presentation and some of the data. These have been diligently and consistently addressed by the authors so that I now see the manuscript as suitable for publication in PLoS Genetics. Most of these related to clarity of presentation and data support, but the revised manuscript also adds interesting new data on an SSA model and enhanced acetylation studies. The study still relies at points on complicated interpretations of multiply mutated strains, but the analysis is thoughtful and meticulous such that I think the conclusions are solid, in additional to being novel and interesting to those studying DSB repair.

With respect to other reviewers’ comments, Reviewer 1 was most negative. I agree that in some ways the study is weakened by a particular focus on G1 where the exact role of some the protein functions might be questioned, e.g., if they are only evident when other proteins are mutated, are they important? However, I tend toward the view that the interactions are reproducible and likely part of the complicated orchestration of protein activities at DSBs in general. As such, perhaps unlike Reviewer 1, I found the overall collection of findings to be coherent and cohesive as drawn in the final model.

**Have all data underlying the figures and results presented in the manuscript been provided?**

Reviewer #1: None

Reviewer #2: Yes

Reviewer #3: Yes

PLOS authors have the option to publish the peer review history of their article (what does this mean?). If published, this will include your full peer review and any attached files.

Reviewer #1: No

Reviewer #2: No

Reviewer #3: No

---

## [Editor Report · Decision Letter 2]

9 Sep 2021

Dear Dr Côté,

We are pleased to inform you that your manuscript entitled "Antagonistic relationship of NuA4 with the Non-Homologous End-Joining machinery at DNA damage sites" has been editorially accepted for publication in PLOS Genetics. Congratulations!

Yours sincerely,

Lorraine S. Symington

Associate Editor

PLOS Genetics

Gregory P. Copenhaver

Editor-in-Chief

PLOS Genetics

Comments from the reviewers (if applicable):

**Data Deposition**

http://datadryad.org/submit?journalID=pgenetics&manu=PGENETICS-D-21-00169R2

**Press Queries**

---

## [Editor Report · Acceptance letter]

15 Sep 2021

PGENETICS-D-21-00169R2 

Antagonistic relationship of NuA4 with the Non-Homologous End-Joining machinery at DNA damage sites 

Dear Dr Côté, 

We are pleased to inform you that your manuscript entitled "Antagonistic relationship of NuA4 with the Non-Homologous End-Joining machinery at DNA damage sites" has been formally accepted for publication in PLOS Genetics! Your manuscript is now with our production department and you will be notified of the publication date in due course.

With kind regards,

Andrea Szabo

PLOS Genetics

On behalf of:
